# Febrile Seizures and Respiratory Viruses Determined by Multiplex Polymerase Chain Reaction Test and Clinical Diagnosis

**DOI:** 10.3390/children7110234

**Published:** 2020-11-17

**Authors:** Ji Yoon Han, Seung Beom Han

**Affiliations:** 1Department of Pediatrics, College of Medicine, The Catholic University of Korea, Seoul 06591, Korea; han024@catholic.ac.kr; 2Department of Pediatrics, Daejeon St. Mary’s Hospital, Daejeon 34943, Korea; 3The Vaccine Bio Research Institute, College of Medicine, The Catholic University of Korea, Seoul 06591, Korea

**Keywords:** febrile seizure, respiratory virus, child

## Abstract

Febrile seizure (FS) is a common benign seizure disorder of young children. Although upper respiratory tract infection is the cause of fever in most episodes of FS, studies to identify respiratory viruses using a multiplex polymerase chain reaction (mPCR) test have rarely been performed for children with FS. Medical records of children presenting with FS between January 2015 and December 2019 were retrospectively reviewed. Respiratory viruses identified by a rapid influenza detection test and mPCR test were investigated, and their seasonal distribution and the association between viral identification and seizure characteristics were determined. A total of 607 episodes of FS were analyzed: 81.1% of cases were generalized tonic–clonic seizures, 81.5% occurred within 24 h after fever onset, and 87.3% continued for ≤5 min. Complex FS occurred in 17.5% of FS episodes, and epilepsy was diagnosed in 2.5% of tracked cases. Of the 138 mPCR tests performed in 235 hospitalized episodes of FS, 112 (81.2%) tested positive for respiratory viruses: rhinovirus, enterovirus, adenovirus, and influenza virus were most frequently identified. The identified respiratory viruses showed similar seasonal distributions as were observed in community-acquired respiratory tract infections. The identification of a specific respiratory virus was not significantly associated with seizure characteristics or the development of complex FS. In conclusion, respiratory viruses, showing similar seasonal distributions with community-acquired respiratory tract infections and no significant association with the severity and outcomes of FS, should not be rigorously tested for in children with FS.

## 1. Introduction

Febrile seizure (FS) is defined as a seizure accompanied by fever, which is not caused by a central nervous system (CNS) infection or metabolic imbalance in children aged between 6 and 60 months and who have no underlying neurologic diseases [1]. FS affects 2–5% of healthy children, and complex FS, defined as seizure with a duration over 15 min, two or more seizures within 24 h, or focal seizures, is associated with subsequent epilepsy [1]. Genetic epilepsy with febrile seizures plus (GEFS+) is a syndromic autosomal dominant disorder, in which affected family members display various seizure disorders, from simple febrile seizures to more serious phenotypes of epilepsy, and FS may be the initial presentation of GEFS+ [2]. Some mutations of ion channels, such as *SCN1A* and *SCN2A*, are known to be associated with GEFS+ [2], and some mutations of ion channels were also presumed to be associated with sudden unexpected death in epilepsy and cardiac arrhythmia [3,4]. One previous study reported a higher rate of FS in sudden death in childhood cases than in the general population [5]. However, most children with FS have favorable outcomes, and long-term neurodevelopmental complications and sequelae occur very rarely, especially in children with simple FS [1]. Therefore, evaluation and treatment for the cause of fever rather than seizures are in greater need during the acute phase of FS.

Fever in children with FS is more likely to be caused by viral infection than severe bacterial infection, and upper respiratory tract infection (URI) is the most common cause of fever [6,7,8]. The severity and complications of respiratory tract infection can differ according to the type of respiratory virus and early diagnosis and anti-viral therapy for some viruses, such as influenza virus, can be helpful. Therefore, early prediction and identification of respiratory viruses causing fever can be useful for acute care of children with FS. While influenza virus and human herpesvirus type 6 and 7 (HHV-6/7) are known as major causes of FS [9], some studies have reported that adenovirus, parainfluenza virus, enterovirus, and respiratory syncytial virus (RSV) are more frequently identified than influenza virus in children with FS [6,7,10,11]. In most previous studies evaluating the association between respiratory viruses and FS, viral culture and antigen detection tests were used to identify respiratory viruses. Since the 2000s, a multiplex polymerase chain reaction (mPCR) test, which has higher sensitivity for identifying respiratory viruses than the mentioned conventional methods, has been widely used in real-life clinical settings [12]. Moreover, there have been few studies using mPCR tests to identify respiratory viruses in children with FS [13,14,15]. We aimed to evaluate the seizure characteristics and cause of fever based on clinical diagnosis and laboratory results in children presenting with FS. In particular, respiratory viruses identified by a rapid influenza detection test (RIDT) and mPCR test were investigated in children with FS.

## 2. Subjects and Methods

Medical records of children, who presented with FS at the Department of Pediatrics or Emergency Medical Center of Daejeon St. Mary’s Hospital (Daejeon, Republic of Korea) between January 2015 and December 2019, were retrospectively reviewed. FS was defined as a seizure accompanied by fever in children aged between 6 and 60 months without underlying neurodevelopmental diseases. Seizures caused by acute CNS infection, other CNS lesions, and metabolic causes, including electrolyte imbalance and hypoglycemia, were excluded from the study. Demographic data, including age and sex, were collected. Seizure characteristics, including type and duration of seizures, number of episodes within 24 h, time interval between seizure occurrence and fever onset, number of previous seizure episodes, family history of FS in 1st degree relatives, results of electroencephalography and brain magnetic resonance images, and the development of subsequent epilepsy were investigated. During the follow-up period after FS, epilepsy was diagnosed when two or more episodes of unprovoked seizures developed, one or more episodes of unprovoked seizures developed with abnormal electroencephalography results, or any type of epileptic syndrome was identified. The clinical diagnosis for fever and accompanying clinical symptoms were also investigated. Serum electrolytes and glucose levels to determine metabolic causes of seizures and blood, urine, and stool culture results to determine pathogenic micro-organisms for fever were investigated. The results of RIDTs and mPCR tests were collected to determine respiratory viruses identified in children with FS. RIDTs and mPCR tests were performed for nasopharyngeal swab samples using commercial kits: Alere BinaxNOW^®^ Influenza A & B Card (Abbott, IL, USA) for RIDTs and Anyplex™ II RV16 Detection kit (Seegene Inc., Seoul, Korea) for mPCR tests. The commercial mPCR test kit can simultaneously identify 16 types of respiratory virus, including adenovirus, influenza A virus, influenza B virus, parainfluenza virus (types 1, 2, 3, and 4), rhinovirus, bocavirus, coronavirus (229E, OC43, and NL63), enterovirus, human metapneumovirus (HMPV), and RSV (groups A and B). Children who were clinically diagnosed with herpangina or hand, foot, and mouth disease (HFMD) were additionally categorized as enterovirus infection, regardless of their mPCR test results. Although laboratory tests for HHV-6/7 were not performed, children diagnosed with exanthem subitum, most of which are caused by HHV-6/7 infection, were categorized as HHV-6/7 infection. Four seasons of the year were defined as follows: spring, March to May; summer, June to August; autumn, September to November; winter, December to February, and seasonal distributions of the identified respiratory viruses were determined. Seizure characteristics were compared between children with positive and negative results for each respiratory virus to determine their effect on the severity and outcome of FS. Since most of the mPCR tests were performed for hospitalized children, clinical and seizure characteristics were compared between hospitalized and non-hospitalized (outpatient department (OPD) or emergency room care) episodes of FS.

In comparisons between non-hospitalized and hospitalized episodes and between children with positive and negative results for each respiratory virus, Kruskal–Wallis and chi-square tests were conducted for continuous and categorical variables, respectively. Statistical analyses were performed using SPSS 21 (IBM Corporation, Armonk, NY, USA), and statistical significance was defined as a *p* < 0.05. This study was approved by the Institutional Review Board of Daejeon St. Mary’s Hospital with a waiver of receiving informed consent (approval number: DC20RASI0057).

## 3. Results

During the study period, 695 episodes of seizures accompanying fever were identified. Among them, 60 (8.6%) episodes in children aged >60 months and 10 (1.6%) episodes in children with underlying neurodevelopmental diseases were excluded from this study. There were no episodes excluded due to CNS infection, severe hyponatremia (serum sodium level < 125 mEq/L), severe hypocalcemia (serum calcium level < 7.0 mg/dL), and hypoglycemia (serum glucose level < 55 mg/dL). After further exclusion of 18 (2.6%) episodes with insufficient records of seizure characteristics, the remaining 607 episodes of FS that occurred in 464 children were eventually included in this study. During the study period, one, four, six, 18, and 68 children experienced FS six, five, four, three, and two times, respectively.

### 3.1. Demographic and Clinical Characteristics of Children with FS

Among the included 607 episodes of FS, 372 (61.3%, one at the OPD and 371 at the emergency room) were categorized into the non-hospitalized group and 235 (38.7%) were categorized into the hospitalized group (Table 1). Generalized tonic–clonic seizures (*n* = 492, 81.1%) were the most common type of FS, of which 81.5% occurred within 24 h after fever onset and 87.3% continued for 5 min or less. Complex FS occurred in 106 (17.5%) episodes, and status epilepticus occurred in eight (1.3%) episodes. There were no children who received intensive care or died from seizures. Excluding 163 (26.9%) episodes in which no follow-up visits at the OPD were performed, the remaining 444 (73.1%) episodes were tracked for a median of 13 months (interquartile range: 2–29 month) after FS. Subsequent epilepsy was diagnosed in 11 (2.5%) patients. The hospitalized group showed longer durations of seizures and more episodes of seizures within 24 h, and eventually a higher frequency of complex FS than the non-hospitalized group (Table 1). Higher frequencies for family history of FS and subsequent epilepsy were observed in the hospitalized group than in the non-hospitalized group (Table 1).

Respiratory symptoms, including cough, rhinorrhea, and sputum, were accompanied in 450 (74.1%) of FS episodes, and URI (*n* = 459, 75.6%) was the most common cause of fever (Table 2). Higher frequencies of lower respiratory tract infection (LRI), acute gastroenteritis associated with gastrointestinal symptoms, and exanthem subitum associated with skin rash were observed in the hospitalized group than in the non-hospitalized group (Table 2).

### 3.2. Identification of Respiratory Viruses in Children with FS

RIDTs and mPCR tests were performed in 257 (42.3%) and 140 (23.1%) episodes, respectively. RIDTs were performed at similar rates in the hospitalized and non-hospitalized groups, whereas almost all of the mPCR tests (98.6%, *n* = 138) were performed in the hospitalized group (Table 2). Therefore, further analyses for respiratory viruses were performed only with the hospitalized group.

In the hospitalized group, RIDTs, mPCR tests, and both tests were performed in 104, 138, and 64 episodes, respectively: RIDTs or mPCR tests were performed in a total of 178 (75.7%) episodes. At least one respiratory virus was identified by an mPCR test in 112 (81.2%) episodes, and 55 (49.1%) of these were positive for two or more viruses (Table 3). Rhinovirus (*n* = 64) was most frequently identified by the mPCR test, followed by adenovirus (*n* = 31) and enterovirus (*n* = 26, Table 3). Thirteen episodes with clinical diagnoses of HFMD or herpangina (mPCR tests were not performed in seven episodes and mPCR tests were negative for enterovirus in six episodes) were additionally categorized into enterovirus infection, and ultimately, 39 episodes were regarded as positive for enterovirus. Influenza viruses were identified in a total of 26 episodes: 18 with positive RIDT results (influenza A virus in 12, influenza B virus in five, and both in one) and eight with positive mPCR test results (both influenza A and B viruses in four). A total of 16 episodes with a diagnosis of exanthem subitum were regarded as positive for HHV-6/7. Among the identified viruses, rhinovirus, adenovirus, and HHV-6/7 showed even seasonal distributions (Figure 1). More HMPVs (*p* = 0.004) in the spring, more enteroviruses in the summer (*p* < 0.001), more RSVs in the autumn and winter (*p* < 0.001), and more influenza viruses (*p* < 0.001) and coronaviruses (*p* = 0.028) in the winter were identified than in other seasons (Figure 1). Parainfluenza viruses tended to be identified in the spring, and most bocaviruses were identified in the spring and summer (Figure 1). Seizure characteristics and the frequency of complex FS were not significantly different between episodes with positive and negative mPCR results for rhinovirus, adenovirus, influenza virus, parainfluenza virus, RSV, HMPV, and coronavirus (Table 4, Appendix A). More enteroviruses were identified in children experiencing two or more numbers of FS than those experiencing their 1st FS (*p* = 0.007, Table 4), whereas more HHVs-6/7 were identified in younger children (*p* < 0.001) experiencing their 1st FS (*p* = 0.003) than older children experiencing multiple FS episodes (Appendix A). Influenza virus-positive children tended to be older than influenza-negative children (Table 4, *p* = 0.002).

## 4. Discussion

Seizure characteristics of children with FS in this study were similar to those reported in previous studies. Fever was caused by respiratory tract infection in most episodes of FS, similar to results of previous studies. The identified respiratory viruses showed different seasonal distributions and none of the identified viruses caused significant changes in seizure characteristics.

Most children in this study experienced a single episode of generalized tonic–clonic type FS, which occurred within 24 h after fever onset and continued for 5 min or less. Complex FS occurred in 17.5% of FS episodes, and epilepsy was diagnosed in 2.5% of tracked cases. These characteristics of FS were similar to those reported by previous studies [6,7,10,11,15,16]. Although epilepsy was subsequently diagnosed in a small portion of FS children, FS may be the initial presentation of epilepsy with genetic causes, which has the potential of sudden death [2,3,4]. Thorough follow-up for a subsequent development of epilepsy and appropriate genetic studies for epilepsy syndromes should be emphasized. URI was the cause of fever in more than 80% of FS episodes, and bacterial infection definitely requiring antibiotic therapy, such as urinary tract infection and occult bacteremia, caused fever in less than 1% of FS episodes. Considering that there were no children who received intensive care or died from FS, favorable outcomes of FS were re-confirmed in this study.

In this study, rhinovirus, adenovirus, and enterovirus were most frequently identified by the mPCR test in children with FS. Although HHV-6/7 is known to be a major cause of FS [9], a previous meta-analysis and recent study using a PCR test to identify HHV-6 did not show a significant association between HHV-6 and FS [17,18]. The low number of episodes with a diagnosis of exanthem subitum in this study seems not to support a higher risk for FS due to HHV-6/7 than other respiratory viruses. Some previous studies have reported influenza virus as the most common cause of FS [10,16,19,20]; however, rhinovirus, adenovirus, and enterovirus were more frequently identified than influenza virus in this study. Moreover, seasonal distributions of identified respiratory viruses differed from one another. Rhinovirus and adenovirus were distributed evenly throughout the year, whereas enterovirus and influenza virus were more frequently identified in the summer and winter, respectively, than in other seasons. Parainfluenza virus was mostly identified from the spring to the autumn, bocavirus was mostly identified in the spring and summer, HMPV was mostly identified in the spring, RSV was identified exclusively in the autumn and winter, and coronavirus was identified mostly in the winter. These seasonal distributions of respiratory viruses were the same as those observed in community-acquired respiratory tract infection in Korea [21]. Previous studies have reported similar seasonal distributions of respiratory viruses between children with FS and those with community-acquired respiratory tract infection [20,22]. Therefore, the discrepancy in frequently identified viruses in children with FS among studies seems not to be caused by the difference in neuro-tropism of a specific respiratory virus, but to be caused by the epidemic extent of each virus during the study period [10]. This study showed no significant association between the identification of a specific respiratory virus and the severity and outcome of FS, in agreement with the findings of previous studies [6,7,10,11,20]. Epilepsy was not associated with specific respiratory viral infection, suggesting that a specific respiratory virus does not provoke FS in children with underlying genetic causes of epilepsy, such as GEFS+. Therefore, testing for respiratory viruses in children with FS should not be required, and causative viruses can be expected based on the regional and seasonal epidemiology of respiratory viral infection. However, testing for influenza virus can be considered during an epidemic of influenza because early anti-viral therapy could be helpful for rapid improvement of influenza symptoms.

Almost all of the mPCR tests in this study were performed in the hospitalized group. Considering the different frequencies of LRI and exanthem subitum between the hospitalized and non-hospitalized groups, HHV-6/7 and viruses capable of causing LRI, including parainfluenza virus, RSV, and HMPV, might be over-identified in this study. However, the seasonal viral distributions observed in this study were similar to those of the Korean community, and the seasonal occurrence rates of FS were comparable between the hospitalized and non-hospitalized groups. The effects of different distributions of clinical diagnoses between the hospitalized and non-hospitalized groups on the distributions of respiratory viruses were expected to be marginal. One study reported similar distributions of respiratory viruses between non-hospitalized and hospitalized children with FS [15], although most previous studies were performed with only hospitalized children with FS.

This study had some limitations. Due to its retrospective nature, there were missing records of seizure characteristics in some episodes. In this study, 26.9% of the included children were lost to follow-up after their FS episodes. Epilepsy might develop in more children than reported in this study, and some of them might be consistent with GEFS+. However, targeted gene panel sequencing for *SCN1A, SCN2A*, and *GABRG2*, which are associated with GEFS+, showed no pathogenic mutations in the 11 children subsequently diagnosed with epilepsy in this study. Because almost all of the mPCR tests were performed in the hospitalized group, a selection bias could not be ignored. Although the mPCR test has higher sensitivity for identifying viruses than conventional methods, colonized or non-viable viruses can generate false positive results and increase co-identification rates of multiple viruses [12]. In this study, multiple viruses were identified in 49.1% of episodes in which mPCR tests were performed, and which one of the co-identified viruses was causative for fever could not be accurately determined. To overcome this intrinsic drawback of the mPCR test, febrile children without seizures and afebrile children with seizures should be considered as control groups. Considering that a specific virus, such as rhinovirus, adenovirus, or RSV, showed discrepant results for an association with FS in different studies [10,11,16,19], a well-designed case–control study should be performed.

## 5. Conclusions

In conclusion, seasonal distributions of respiratory viruses identified in children with FS were similar to those of community-acquired respiratory tract infection, and identification of a specific respiratory virus was not significantly associated with the severity and outcome of FS. Therefore, viral tests for children with FS should not be rigorously performed, except for influenza virus, for which early anti-viral therapy could be helpful.

## Figures and Tables

**Figure 1 children-07-00234-f001:**
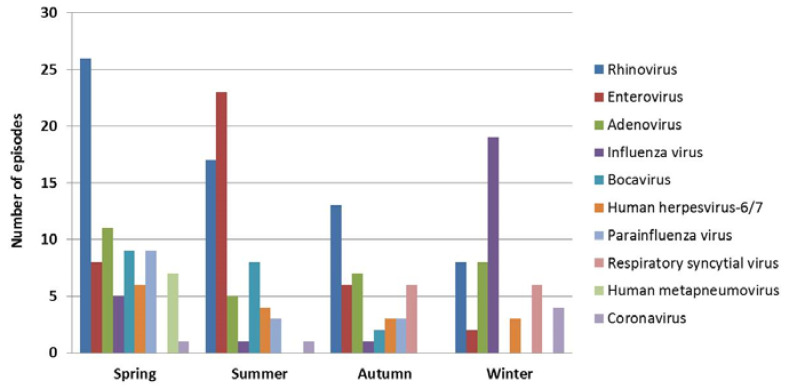
Seasonal distributions of respiratory viruses in children with febrile seizures.

**Table 1 children-07-00234-t001:** Demographic and seizure characteristics of children presented with febrile seizures.

Factor	Non-Hospitalized Group (*n* = 372)	Hospitalized Group (*n* = 235)	*p*-Value
Age, months, median (IQR)	24 (17–35)	24 (16–34)	0.351
Male sex	219 (58.9)	149 (63.4)	0.265
Classification of febrile seizure			<0.001
Simple febrile seizure	354 (95.2)	147 (62.6)	
Complex febrile seizure	18 (4.8)	88 (37.4)	
>15 min of duration	2 (11.1)	16 (18.2)	
Recur within 24 h	14 (77.8)	71 (80.7)	
Focal seizures	2 (11.1)	0 (0.0)	
Type of seizure			0.588
Generalized tonic–clonic	298 (80.1)	194 (82.6)	
Generalized tonic	42 (11.3)	26 (11.1)	
Focal	2 (0.5)	0 (0.0)	
Others	30 (8.1)	15 (6.4)	
Duration of seizure			<0.001
≤5 min	339 (91.1)	191 (81.3)	
≤15 min	31 (8.3)	28 (11.9)	
≤30 min	1 (0.3)	9 (3.8)	
>30 min	1 (0.3)	7 (3.0)	
Number of episodes within 24 h			<0.001
1	358 (96.2)	164 (69.8)	
2	13 (3.5)	60 (25.5)	
≥3	1 (0.3)	11 (4.7)	
Number of seizure attacks			<0.001
1	215 (57.8)	106 (45.1)	
2	80 (21.5)	43 (18.3)	
3–5	69 (18.5)	65 (27.7)	
>5	8 (2.2)	21 (8.9)	
Time interval between seizure occurrence and fever onset ^a^			0.134
Concurrent	56 (15.2)	24 (10.3)	
<6 h	100 (27.2)	51 (22.0)	
<12 h	78 (21.2)	56 (24.1)	
<24 h	69 (18.8)	61 (26.3)	
<72 h	61 (16.6)	37 (15.9)	
≥72 h	4 (1.1)	3 (1.3)	
Abnormal EEG result ^b^	0 (0.0)	7 (7.2)	1.000
Abnormal brain MRI result ^c^	0 (0.0)	2 (3.5)	1.000
Family history of febrile seizure	75 (20.2)	76 (32.3)	0.001
Subsequent diagnosis of epilepsy	2 (0.5)	9 (3.8)	0.004

IQR: interquartile range; EEG: electroencephalography; MRI: magnetic resonance image. ^a^ Determined in 368 episodes of the non-hospitalized group and in 232 episodes of the hospitalized group. ^b^ Determined in six episodes of the non-hospitalized group and in 97 episodes of the hospitalized group. ^c^ Determined in one episode of the non-hospitalized group and in 57 episodes of the hospitalized group.

**Table 2 children-07-00234-t002:** Clinical symptoms and causes of fever in children presented with febrile seizures.

Factor	Non-Hospitalized Group (*n* = 372)	Hospitalized Group (*n* = 235)	*p*-Value
Clinical symptoms			
Cough	223 (59.9)	150 (63.8)	0.338
Rhinorrhea	247 (66.4)	156 (66.4)	0.997
Sputum	141 (37.9)	102 (43.4)	0.178
Dyspnea	1 (0.3)	2 (0.9)	0.563
Vomiting	19 (5.1)	22 (9.4)	0.042
Diarrhea	14 (3.8)	28 (11.9)	<0.001
Skin rash	5 (1.3)	23 (9.8)	<0.001
Cause of fever			<0.001
Upper respiratory tract infection	321 (86.3)	138 (58.7)	
Lower respiratory tract infection	16 (4.3)	30 (12.8)	
Hand, foot, and mouth disease/herpangina	16 (4.3)	24 (10.2)	
Acute gastroenteritis	6 (1.6)	18 (7.7)	
Fever without a focus	11 (3.0)	6 (2.6)	
Exanthem subitum	0 (0.0)	16 (6.8)	
Urinary tract infection	1 (0.3)	1 (0.4)	
Occult bacteremia	1 (0.3)	1 (0.4)	
Kawasaki disease	0 (0.0)	1 (0.4)	
Season			0.393
Spring	102 (27.4)	79 (33.6)	
Summer	93 (25.0)	58 (24.7)	
Autumn	63 (16.9)	36 (15.3)	
Winter	114 (30.6)	62 (26.4)	
Performed viral tests			
Rapid influenza detection test	153 (41.1)	104 (44.3)	0.448
Multiplex polymerase chain reaction test	2 (0.5)	138 (58.7)	<0.001

**Table 3 children-07-00234-t003:** Results of multiplex polymerase chain reaction tests for respiratory viruses.

Viruses	Number (%)
Rhinovirus	23 (20.5)
Adenovirus	6 (5.4)
Enterovirus	7 (6.3)
Parainfluenza virus	4 (3.6)
Bocavirus	3 (2.7)
Respiratory syncytial virus	5 (4.5)
Influenza virus (A/B)	5 (2/3) (4.5)
human metapneumovirus	2 (1.8)
Coronavirus	2 (1.8)
Rhinovirus + adenovirus	7 (6.3)
Rhinovirus + enterovirus	8 (7.1)
Rhinovirus + parainfluenza virus	5 (4.5)
Rhinovirus + bocavirus	1 (0.9)
Rhinovirus + respiratory syncytial virus	1 (0.9)
Rhinovirus + influenza B virus	1 (0.9)
Rhinovirus + human metapneumovirus	2 (1.8)
Adenovirus + enterovirus	3 (2.7)
Adenovirus + coronavirus	1 (0.9)
Enterovirus + bocavirus	1 (0.9)
Enterovirus + respiratory syncytial virus	1 (0.9)
Parainfluenza virus + bocavirus	2 (1.8)
Respiratory syncytial virus + coronavirus	1 (0.9)
Influenza A virus + human metapneumovirus	1 (0.9)
Rhinovirus + adenovirus + enterovirus	1 (0.9)
Rhinovirus + adenovirus + parainfluenza virus	2 (1.8)
Rhinovirus + adenovirus + bocavirus	4 (3.6)
Rhinovirus + adenovirus + respiratory syncytial virus	2 (1.8)
Rhinovirus + adenovirus + influenza A virus	1 (0.9)
Rhinovirus + adenovirus + coronavirus	1 (0.9)
Rhinovirus + enterovirus + bocavirus	1 (0.9)
Rhinovirus + enterovirus + respiratory syncytial virus	1 (0.9)
Rhinovirus + parainfluenza virus + bocavirus	1 (0.9)
Rhinovirus + bocavirus + respiratory syncytial virus	1 (0.9)
Adenovirus + bocavirus + human metapneumovirus	2 (1.8)
Enterovirus + parainfluenza virus + bocavirus	1 (0.9)
Enterovirus + bocavirus + coronavirus	1 (0.9)
Rhinovirus + adenovirus + enterovirus + bocavirus	1 (0.9)

**Table 4 children-07-00234-t004:** Comparisons between children with positive and negative results for each respiratory virus.

Factor	Rhinovirus			Enterovirus			Adenovirus			Influenza Virus		
	Negative (*n* = 74)	Positive (*n* = 64)	*p*-Value	Negative (*n* = 196)	Positive (*n* = 39)	*p*-Value	Negative (*n* = 107)	Positive (*n* = 31)	*p*-Value	Negative (*n* = 152)	Positive (*n* = 26)	*p*-Value
Age, months, median (IQR)	22 (14–31)	23 (16–33)	0.773	22 (15–34)	27 (21–36)	0.055	22 (15–30)	22 (15–33)	0.884	22 (15–31)	35 (23–41)	0.002
Male sex	44 (59.5)	42 (65.6)	0.456	123 (62.8)	26 (66.7)	0.643	65 (60.7)	21 (67.7)	0.479	94 (61.8)	19 (73.1)	0.272
Complex febrile seizure	28 (37.8)	27 (42.2)	0.603	75 (38.3)	13 (33.3)	0.561	43 (40.2)	12 (38.7)	0.882	60 (39.5)	6 (23.1)	0.110
Type of seizure			0.253			0.387			0.179			0.847
Generalized tonic–clonic	64 (86.5)	53 (82.8)		159 (81.1)	35 (89.7)		91 (85.0)	26 (83.9)		127 (83.6)	22 (84.6)	
Generalized tonic	5 (6.8)	9 (14.1)		24 (12.2)	2 (5.1)		9 (8.4)	5 (16.1)		15 (9.9)	3 (11.5)	
Others	5 (6.8)	2 (3.1)		13 (6.6)	2 (5.1)		7 (6.5)	0 (0.0)		10 (6.6)	1 (3.8)	
Duration of seizure			0.702			0.921			0.060			0.309
≤5 min	58 (78.4)	51 (79.7)		159 (81.1)	32 (82.1)		89 (83.2)	20 (64.5)		120 (78.9)	23 (88.5)	
≤15 min	11 (14.9)	7 (10.9)		24 (12.2)	4 (10.3)		12 (11.2)	6 (19.4)		20 (13.2)	3 (11.5)	
>15 min	5 (6.8)	6 (9.4)		13 (6.6)	3 (7.7)		6 (5.6)	5 (16.1)		12 (7.9)	0 (0.0)	
Number of episodes within 24 h			0.977			0.924			0.761			0.684
1	50 (67.6)	43 (67.2)		136 (69.4)	28 (71.8)		71 (66.4)	22 (71.0)		104 (68.4)	20 (76.9)	
2	20 (27.0)	17 (26.6)		51 (26.0)	9 (23.1)		29 (27.1)	8 (25.8)		40 (26.3)	5 (19.2)	
≥3	4 (5.4)	4 (6.3)		9 (4.6)	2 (5.1)		7 (6.5)	1 (3.2)		8 (5.3)	1 (3.8)	
Number of seizure attacks			0.541			0.007			0.197			0.090
1	32 (43.2)	31 (48.4)		96 (49.0)	10 (25.6)		52 (48.6)	11 (35.5)		74 (48.7)	8 (30.8)	
≥2	42 (56.8)	33 (51.6)		100 (51.0)	29 (74.4)		55 (51.4)	20 (64.5)		78 (51.3)	18 (69.2)	
Time interval between seizure occurrence and fever onset ^a^			0.431			0.258			0.934			0.497
Concurrent	8 (11.0)	8 (12.9)		17 (8.8)	7 (18.4)		13 (12.4)	3 (10.0)		18 (12.1)	1 (3.8)	
<24 h	48 (65.8)	44 (71.0)		144 (74.2)	24 (63.2)		71 (67.6)	21 (70.0)		105 (70.5)	19 (73.1)	
<72 h	17 (23.3)	9 (14.5)		31 (16.0)	6 (15.8)		20 (19.0)	6 (20.0)		24 (16.1)	6 (23.1)	
≥72 h	0 (0.0)	1 (1.6)		2 (1.0)	1 (2.6)		1 (1.0)	0 (0.0)		2 (1.3)	0 (0.0)	
Abnormal EEG result	1/30 (3.3)	2/27 (7.4)	0.595	5/76 (6.6)	2/15 (13.3)	0.325	2/40 (4.8)	1/15 (6.7)	0.526	3/63 (4.8)	0/9 (0.0)	1.000
Abnormal brain MRI result	0/19 (0.0)	1/11 (9.1)	0.367	2/49 (4.1)	0/8 (0.0)	1.000	0/23 (0.0)	1/7 (14.3)	0.233	1/35 (2.9)	0/3 (0.0)	1.000
Family history of febrile seizures	26 (35.1)	19 (29.7)	0.496	59 (30.1)	17 (43.6)	0.100	35 (32.7)	10 (32.3)	0.962	50 (32.9)	5 (19.2)	0.164
Subsequent diagnosis of epilepsy	4 (5.4)	2 (3.1)	0.686	8 (4.1)	1 (2.6)	1.000	4 (3.7)	2 (6.5)	0.616	6 (3.9)	0 (0.0)	0.595

IQR: interquartile range; EEG: electroencephalography; MRI: magnetic resonance image. ^a^ Determined in 135 episodes.

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
