# Peer review of "Febrile Seizures and Respiratory Viruses Determined by Multiplex Polymerase Chain Reaction Test and Clinical Diagnosis"

_children, 2020, doi:10.3390/children7110234_

Round 1

Reviewer 1 Report

In this study by Han & Han, they investigated the presence of seasonal respiratory viruses in children with febrile seizures, with more in-depth PCR data from the subset of patients who were hospitalized. Importantly, they found that of the various respiratory viruses present in the patients, none of these had any predictive value in terms of developing subsequent epilepsy or other important sequelae.

My one suggestion would be to use a correction for multiple comparisons in Table 4. Currently, any comparison with a p < 0.05 is reported as significant; however, across 88 comparisons, one would expect for there to be spurious "significant" comparisons at p < 0.05. The simplest correction would be Bonferroni (so, p < 0.05/88 or 0.0006). This would not change your conclusions or discussion, it would just make this table's results a little easier to interpret. 

Author Response

We thank you for your kind comments. 

Reviewer 2 Report

The authors try to propose the study on the significance of molecular testing for respiratory viruses in order to make a physipathologic correlation with  Febrile seizure (FS), which is  common benign seizure disorder of young children.

The introduction needs to be expanded in order to give to the reader the overview on FE, also from a genetic point of view. It is well know that genetic epilepsy with febrile seizures plus (GEFS+) is worth of investigation.

I would recommend these references in order to improve the background:

  • Genetic literacy series: genetic epilepsy with febrile seizures plus Kenneth A MyersIngrid E Scheffer Samuel F Berkovic ILAE Genetics Commission,. Epileptic disord 2018 Aug 1;20(4):232-238. doi: 10.1684/epd.2018.0985.

  • Genetic investigation of sudden unexpected death in epilepsy cohort by panel target resequencing,  Coll et al. results section
    i do completely agree regarding the interpretation of the results especially on the lack of causal proof between respiratory viruses and FE. I do suggest the authors to include a limitation of the study section in order to highlight the weakness of the paper (for instance the absence of genetic investigation in the cohort).  

Author Response

We thank you for your valuable advice.

Round 2

Reviewer 2 Report

The Authors have fulfilled partially the requested modifications.

There is still missing reference that has been indicated in the previous indications (first round of review) that the authors did not included: these 2 papers

  • Coll M, Allegue C, Partemi S, et al Genetic investigation of sudden
    unexpected death in epilepsy cohort by panel target resequencing. Int J Legal
    Med. 2016 Mar;130(2):331-9. doi: 10.1007/s00414-015-1269-0. Epub 2015 Sep 30.
    PMID: 26423924.
  • Partemi S, Vidal MC, Striano P,  et al. Genetic and forensic implications in epilepsy and cardiac arrhythmias: a case series. Int J Legal Med. 2015 May;129(3):495-504. doi: 10.1007/s00414-014-1063-4. Epub 2014 Aug 15. PMID: 25119684.

are the proof of concept that some of FS (febrile seizures) are observed in epilepsy event provoking is some cases  sudden death (SUDEP). These part of the literature should be acknowledge. It is a reasonable information that should be included in this paper, as far as some of these events (FS) mandates further research into the potential link between simple febrile seizures and death in children (sudden unexplained death in childhood (SUDC) is the sudden death of a child older than 1 year) . 

Author Response

We thank you for your kind advice, again. 

Please see the attached file for our response.
